# Specific Learning Disorder in Children and Adolescents, a Scoping Review on Motor Impairments and Their Potential Impacts

**DOI:** 10.3390/children9060892

**Published:** 2022-06-15

**Authors:** Mariève Blanchet, Christine Assaiante

**Affiliations:** 1Laboratoire de Recherche en Motricité de L’enfant, Département des Sciences de L’activité Physique, Université du Québec à Montréal, 141 Av. Président-Kennedy, Montréal, QC H2X 1Y4, Canada; 2LNC, UMR 7291, Fédération 3C, AMU-CNRS, Centre Saint-Charles, Pole 3C, Case C, 3 Place Victor Hugo, 13331 Marseille, France; christine.assaiante@univ-amu.fr

**Keywords:** learning disabilities, motor development, sensorimotor representations, locomotion, posture, gross motor skills, fine motor skills, children and adolescents

## Abstract

Mastering motor skills is important for children to achieve functional mobility and participate in daily activities. Some studies have identified that students with specific learning disorders (SLD) could have impaired motor skills; however, this postulate and the potential impacts remain unclear. The purpose of the scoping review was to evaluate if SLD children have motor impairments and examine the possible factors that could interfere with this assumption. The sub-objective was to investigate the state of knowledge on the lifestyle behavior and physical fitness of participants with SLD and to discuss possible links with their motor skills. Our scoping review included preregistration numbers and the redaction conformed with the PRISMA guidelines. A total of 34 studies published between 1990 and 2022 were identified. The results of our scoping review reflected that students with SLD have poorer motor skills than their peers. These motor impairments are exacerbated by the complexity of the motor activities and the presence of comorbidities. These results support our sub-objective and highlight the link between motor impairments and the sedentary lifestyle behavior of SLDs. This could lead to deteriorating health and motor skills due to a lack of motor experience, meaning that this is not necessarily a comorbidity. This evidence emphasizes the importance of systematic clinical motor assessments and physical activity adaptations.

## 1. Introduction

Many children have a potentially disabling condition that interferes with their learning abilities at school. Compared to their peers, teachers denote problems related to these children’s reading, delays in their written expression skills and/or impairments in the mathematical domain. For some children, these learning difficulties are temporary and can be corrected with adapted interventions. For other children with learning difficulties, learning skills of a mild enough severity level could be compensated [1]. However, for from 5% to 15% of children, these impairments are persistent and significant, despite the appropriate interventions [1]. These learning disorder disrupt the normal pattern of learning academic skills. In these cases, children have a neurodevelopmental disorder known as learning disabilities, DYS disorders or the DYS constellation; the first part of the disorder name “DYS” (*dys*function) regroups many kinds of learning and developmental problems, such as dysorthographia, dysphasia, dyslexia, dysgraphia, dyscalculia, dysexecutive syndrome and dyspraxia. 

In the latest edition of the Diagnostic and Statistical Manual of Mental Disorders (DSM-5), published in 2013, the characterization of the specific learning disability (SLD) was updated. SLD consists of four key elements: (1) characterized by constant difficulties in learning and using one or more of the academic domains (i.e., reading comprehension, arithmetic computation and/or written expression) for at least six months although target skill interventions have been given; (2) academic skills are below what is expected at the individual’s age, which impairs functioning in school, at work and in activities of daily living; (3) the SLD is diagnosed at the age of onset, at school age, or when higher-level skills are demanded and (4) those who have intellectual developmental disorders, global developmental delays, hearing or vision disorders, psychosocial difficulties, language differences and who lack proficiency in the language of academic instruction are excluded [1]. The DSM-5, as well as many studies, identified that SLDs frequently co-occur with language disorders, speech sound disorders, developmental coordination disorders (DCDs), attention-deficit/hyperactivity disorder (ADHD) and mental disorders (e.g., anxiety disorders, depressive and bipolar disorders). For example, one in three children with ADHD also has a specific learning disability [2].

The diagnosis of SLD is based on a clinical interview. It can also be ascertained from school reports and descriptions of previous educational or psychological assessments. To increase the certainty of the diagnosis, rating scales from standardized tests or subtests within the academic domain, which are recognized and valid in the country in which the tests are performed, have been recommended [1]. However, there is another notable clinical sign of SLDs [3], which is not systematically associated with DCD but has been reported by many studies, although most of the time it is not evaluated: this is poor motor skills, without an associated complaint. 

Although the motor difficulties in this population are unrecognized in the school and sports systems, it has been 65 years since the first studies observed, measured and indicated motor control impairments in children and adolescents with learning disorders [4,5,6,7]. These children have been reported to display impairments in postural, gross and fine motor skills relative to peers without such disabilities. Over the decades, a growing body of the literature has supported these postulates. In 2020, Baharudin and colleagues published a review of five standardized motor tools that assess the movement and function of children with SLD [8]. Additionally, in 2016, a meta-analysis by Rochelle and Talcott investigated the balance of children with dyslexia. These studies have been extensively discussed in the literature [9]. Moreover, several studies have broadened the analyses as well as the interpretations of motor impairments and/or delays in SLDs, notably due to the evolution of motor tests and the characterization of SLDs, as well as co-morbidities such as DCD [8,9,10,11,12,13,14,15]. Some researcher groups have deepened the understanding of the mechanisms that could potentially form part of these motor impairments using neuro-imaging techniques [16,17,18]. Nevertheless, motor difficulties in these populations remain misunderstood and this question is still open. To the best of our knowledge, no systematic reviews or meta-analyses have been conducted to identify possible motor impairments and/or delays in children and adolescents with SLDs.

Mastering fundamental skills is crucial for children to achieve a good performance in terms of functional mobility and participate in daily activities [19,20], especially during sensitive periods of development [21,22,23,24,25]. Many animal and human studies demonstrated this heightened plasticity during sensitive periods of development in childhood and adolescence [22,23,24,25,26]. Moreover, recent developmental fMRI studies [27] have revealed that the neural bases of the body schema, although established from the age of 7, continue to mature based on a functional pruning of the proprioceptive network, starting during early adolescence, and continuing until adulthood. This neuronal plasticity allows for the central nervous system to learn skills and remember the information needed to reorganize neuronal networks in response to environmental stimulation over the lifespan [21]. However, children with poor motor skills develop low self-perceived motor-skill competences, which consequently leads to sedentary behavior [28]. The dynamic and synergistic roles that motor competences play in the decline in participation in physical activities induces suboptimal sensorimotor inputs during these important developmental stages and could limit proper brain development, with lifelong consequences [21,24]. In addition, low motor skills interfere with learning abilities [29,30,31,32] and impact physical fitness levels [28,33]. Therefore, it is essential to first review the state of knowledge on this important topic by looking at the literature on children and adolescents with SLDs, as well determining several impacts that could be underestimated.

The purpose of this scoping review was to investigate if SLD children had motor deficits and examine the possible factors that could interfere with this assumption. This scoping review could reveal more about the body of evidence on this particular topic: fine, gross and balance skills in SLD children. The sub-objective of this scoping review was to investigate the state of knowledge regarding the lifestyle behavior and physical fitness of children and adolescents with SLDs and to discuss its possible links with their motor skills. Stodden and colleagues (2014) proposed a conceptual model that links motor development and health-related fitness during ontogenesis. We explore this broad question and aim to shed light on the associated SLD signs that contribute to the initiation and maintenance of sedentary behavior, a major health problem that is recognized worldwide. The World Health Organization considers sedentary behavior to be a growing risk factor: “Physical inactivity is one of the leading risk factors for noncommunicable diseases and death worldwide” [34]. Since physical and motor activities occupy a very important role in the daily life of children and, more broadly, in well-being for any age, the findings of this scoping review could highlight the need for actions and news studies that would favor social inclusion to ensure and support the development of children and adolescents with SLDs. Understanding the most salient factors that influence health and well-being and how the relationships between these factors change over time is critical for future research in this area.

## 2. Materials and Methods

The redaction of this scoping review conformed to the PRISMA guidelines in the PRISMA extension for scoping reviews of Tricco (2018). The registration number, created on 13 September 2021, 12:14 p.m., was DOI 10.17605/OSF.IO/28R7V. Peer-reviewed primary and secondary studies in children and adolescents that were published between 1 January 1950 and 20 May 2022 were eligible for inclusion. Different types of peer-reviewed review were also included, but books, encyclopedia, abstract, blog articles, and study cases were excluded. This scoping review is the continuation of a previous recension of studies, supported by *table pour un mode de vie physiquement actif* (TMVPA: provincial governmental and non-governmental organizations grouping in Québec, Canada). We published an open-source review of the literature focusing on the science regarding participation in physical and motor activities by children with a learning disability, looking at those that included a section on motor impairments in SLDs. This first clearing of the literature was conducted by three students and a professor. Six months after its publication, the authors of this scoping review conducted a rigorous screening of the published studies that evaluated motor skills in SLDs in Pubmed, Proquest, Scopus (Figure 1). Proquests have a “university publications” filter; we obtained over 60,000 results in this category, compared to only a few dozen in other databases. The records were then screened, first according to the type of publication (books, abstracts, etc.), as well as the study’s title, in order to exclude studies that are completely out of scope. Then, we screened abstracts using web-based text, and researchers discussed all ambiguous abstracts to exclude studies. Subsequently, the full texts of all potential articles were reviewed. The reference sections of the included studies were examined to supplement the database records and identify the studies that were not cited in our first publication. 

The screening process was based on both motor and population eligibility criteria. The criteria for motor parameters included gross and/or fine and/or postural and/or posture and/or balance motor skills. Eligibility criteria for populations were children and/or adolescents (ranging from 4 to 17 years of age) with learning disorders. In 2013, the criteria for learning disorders were updated and they were named specific learning disorders (SLDs). To avoid missing studies due to the evolution of terminologies relating to learning disabilities over time, several terms were used. In agreement with the DSM, the review of Grigorenko and colleagues (2020) and the investigation of the references sections published in the studies, the eligibility criteria related to diagnosis were learning disability.ies, learning disabled, specific learning disability.ies, specific word reading disability.ies (SWRD), specific learning disability.ies with reading impairment, specific reading comprehension disability.ies (SRCD), specific learning disability.ies with written expression impairment, specific written expression disability.ies (SWED), math specific learning disability.ies and specific learning disability.ies in mathematics (SMLD) [35]. Additionally, alternative common terms used to refer to a pattern of learning difficulties, such as dyscalculia, dysorthographia and dyslexia, were used. All these keywords concerning the population were combined in database searches with motor parameter keywords in separate research. This research strategy is illustrated in Figure 1 as the identification stage (all synonyms are included but, to simplify the diagram, only one keyword is represented).

The exclusion criteria for our study were adults (>18 years old), intellectual disabilities (QI < 80), Autism Spectrum Disorder, children with familial risk of SLDs, visual or hearing impairments, developmental delays and neurological or muscular disorders such as cerebral palsy, muscular dystrophy or other orthopedic impairments [1]. However, studies with mixed-age populations or mixed-disorder groups were included if it was possible to extract data from them separately. The samples and the protocol for the motor evaluations used in each study were deeply revised and reported (Figure 1 and Table 1). Particular attention was paid to the presence of certain comorbidities in the samples that could alter motor performance, such as AHDH and DCD. However, although the first group recognized congenital maladroitness in 1925, it was only in 1987 that the DSM-III-R (American Psychiatric Association, 1987) included a separate entry for children with developmental perceptual–motor problems and, in 1994, a distinct movement skills syndrome, which the American Psychiatric Association and the World Health Organization classified as “Developmental Coordination Disorder” (DCD) [1,36]. Independent criteria for the assignment of the label DCD were shown to be sadly lacking in 1998 [37]. Thus, this scoping review included studies from 1987, but focused on studies published from 2000 to 2022 in order to minimize confusion in various diagnostics. If the study sample constituted only those with developmental coordination disorder, dyspraxia or ADHD populations, this study was excluded. We found 36 studies who met the inclusion criteria: 17 studies in which investigators did not achieve or reported adequate comorbid disorders’ exclusion and 19 studies in which investigators excluded comorbidities (*n* = 17) or controlled for them in their analyses (*n* = 2) (Table 1).

## 3. Results

The results of this scoping review first report studies with groups that included mixed learning disabilities. Then, the three types of SLD are presented: dyslexia (SWRD, SRCD), dyscalculia (SMLD) and dysorthographia (SWED). Each of these sections presented three movement categories (fine, gross and postural skills), assessed by different quantitative and qualitative motor tests [69], such as the first [70] and second edition [71] of the Test of Gross Motor Development (TGMD) [11,12,44], the first [6] and second editions of the Bruininks–Oseretsky Test of Motor (BOT) [38,56,72,73], the first and second version of Movement Assessment Battery for Children (M-ABC) [19,37,43,44,47,48,68], the Functional Mobility domain from the Pediatric Evaluation of Disability Inventory-Computer Adaptive Test (PEDI-CAT) [19], Spiral Drawing Test [15], Beery–Buktenica Developmental Test of Visual–Motor Integration [68], Leonard Tapping Task [45], Dysgraphia Scale [74], battery of clinical cerebellar tests [60,75], quiet standing paradigms [46,49,50,51,52,53,54,57,62,67], postural stability limits [41], Sensory Organisation Test [40], Multitest Equilibre [55,58], TechnoConcept^®^ platform [59,61], Motor Coordination Test [40], Chinese Handwriting Assessment Tool [63], Dyslexia Screening Test [66], tapping in time to an entraining metronome with fingers [65], and the bead-threading and peg-board tasks [64]. The design of these paradigms allowed for impairments in various sensory systems (visual, vestibular, proprioception, tactile) to be investigated, as well as multisensory integration for movement and postural control. These inputs are a very important part of movement and postural control and several studies have hypothesized that they could possibly be linked with motor skill impairments in SLDs [10,46,54,59,67,76,77]. However, to simplify the first screening of motor skills in children and adolescents with SLDs, our scoping review mainly reported results in the eyes-open condition. These motor assessments results are followed by a brief overview of the literature concerning the physical condition of children and adolescents with SLDs. 

The studies included in our scoping review and published after 1987 were carried out in a total of 16 countries. They were mainly conducted in Brazil (*n* = 7) and France (*n* = 6), but also in Canada (Québec, Ontario and Alberta) (*n* = 4), Netherlands (*n* = 4), United Kingdom (*n* = 3), Norway (*n* = 2), the United States (Indiana and New Castle) (*n* = 2), Germany (*n* = 1), Belgium (*n* = 1), Malesia (*n* = 1), Italia (*n* = 1), China (*n* = 1), Mexico (*n* = 1), Pakistan (*n* = 1), Tunisia (*n* = 1), Egypt (*n* = 1).

### 3.1. Motor Assessment Results in Mixed Group of Learning Disabilities

#### 3.1.1. Standard Qualitative Motor Assessment Batteries in Mixed Group of Learning Disabilities 

Before 1994, many studies reported that children with SLDs had poor motor efficiency [4,5,7,78,79]. For example, in 1977, the motor assessment battery created by Bruininks and Bruininks showed that American children aged from 6 to 13 years with learning disorders had a significantly lower performance on measures of fine, gross, and balance motor skills when the performance was assessed using the Bruininks–Oseretsky Test of Motor (BOT). The results of this study showed that, except for proficiency in the speed of response to a moving visual stimulus, SLD children showed a significantly poorer performance compared to nondisabled students in all motor tasks (running speed and agility, static standing balance, bilateral coordination, strength during sit-ups, push-ups, and standing broad jump, hand–eye coordination ability with gross and fine motor tasks, speed and dexterity during manipulative tasks). However, it is important to note that the specific DCD disorder was not recognized. After the prompted recognition of DCD, Bluechardt and Shephard (1996) used the same motor test as previous researchers to compare the motor performance of 20 children with SLDs from the United States (Indiana), aged from 6 to 8 years, to the North American population norms. Their results supported those of Bruininks and Bruininks (1977) and revealed that the motor performance scores for children with SLDs were significantly reduced, even in the results concerning the speed of response to a visual stimulus [72]. The children were matched according to age, sex and their school officials. However, the authors did not reveal whether children with comorbidities were excluded from the study. In the same vein, Hussein and colleagues (2020) used the second edition of the BOT with SLDs (no information was given on DCD and ADHD) and typical developmental Egyptian children aged from 9 to 13 years. The results demonstrated significant differences in the standard scores for fine, gross, and total motor composites. The SLD group had difficulties regarding fine motor composite as follows: 96% were below the average level of fine motor precision and fine motor integration, 42% below average in terms of manual dexterity, and 98% below average in upper-limb coordination. The observation of gross motor composites showed that 80% of children with SLD were below average in terms of bilateral coordination, 58% were below average in terms of balance, 74% were below average in running speed and agility, and 68% were below average in terms of strength (knee push-ups) [38]. These results are consistent with the previously reported results of Okuda and Pinheiroa (2015) that show inferior BOT-2 scores performance in a Brazilian SLD group (aged from 8 to 11 years) for motor areas and motor subtests when compared with their peers. These authors observed statistically significant differences between the groups regarding scores for fine motor integration, balance, and running speed and agility [42].

These results were also reported with another motor test, the TGMD-2 [71], which qualitatively measured 12 gross motor skills that were divided into locomotor skills (run, gallop, hop, leap, jump, and slide) and object-control skills (two-hand strike, stationary bounce, catch, kick, overhand throw, and underhand roll). The study of Westendorp and colleagues (2011) revealed that SLD children (those with attention deficit/hyperactivity disorder and autism spectrum disorders were excluded but no information was given about other neurological or sensorial disorders) in primary special-needs schools in northern regions of the Netherlands obtained significantly lower scores on both subtests compared to children with typical development. Moreover, the longitudinal study of this cohort indicated a significant gap between children with SLDs and typically developing peers at all ages, i.e., between 7 and 11 years old. The large difference in ball skills between both groups is notable at 7 years, while the difference between both groups at age 11 is smaller. In other words, children with SLDs develop ball skills later in the primary school period compared to their typically developing peers. In contrast, between-group comparisons for locomotor skills showed that children with SLDs scored lower than typically developing children at all ages, except at the younger age (7 years old) [12]. Their sample of 56 children included 15 children with comorbid disorders (9 with ADHD, 3 with autism spectrum disorders, and 3 children diagnosed with both) that were screened by a short medical history. However, these comorbid variables as well as IQ, sex, and age did not contribute significantly to their statistical analysis model. 

At the same time (2011), some members of the Westendorp research team also attempted to deepen the understanding of motor dysfunction in SDL with 137 children aged from 7 to 12 years. This cohort was also recruited from elementary special needs schools in the Netherlands, but was evaluated with another motor test, the M-ABC. Each age band in this test comprised eight items that were divided into three subscales: manual dexterity (bimanual coordination, eye–hand coordination, speed and accuracy of each hand), aiming and catching (catching a moving object, aiming at a goal), and balance (static balance, dynamic balance while moving slowly, dynamic balance while moving fast). Their study indicated that 50.4% of SLD children had significant motor problems, of which 35.0% were definite motor problems (the 5th percentile and below), and 15.4% of SLD children had borderline problems (from the 15th to 6th percentile scores) [39]. However, these scores included 23 children (16.8%) that were diagnosed with comorbid ADHD and 19 children (13.9%) with pervasive developmental disorder not otherwise specified (PDD-NOS). The interaction effects between co-occurring diseases and motor scores were not investigated in the study and no specifications were given regarding DCD inclusion or exclusion. Their results were, therefore, supported in 2003 and 2019 by different research groups. In Malesia, an analysis of total M-ABC-2 scores of 148 SLD children and adolescents aged from 4 to 16 years old revealed that almost half (46%) of the participants had movement dysfunctions (on or below the 15th percentile rank). The exclusion criteria of disorders co-occurring with SLD were unspecified in this study; however, other motor disabilities that possibly included DCD and ADHD were excluded [19]. In 2003, Jongsmans and colleagues used the first edition of M-ABC in the Netherlands and Germany to investigate the motor performance of a typical developmental children group, an SLD group (referred by a LOW special education school; ADHD unspecified) and a DCD group (referred by their physician and meeting the DSM-IV DCD criteria). Among the children without DCD, the mean total M-ABC scores of students with SLDs were significantly poorer than those in the group without SLD. Children with SLDs obtained poorer total M-ABC scores compared to control children, with 32% of children scoring on or below the 15th percentile. The oldest children with SLDs obtained significantly poorer scores compared to younger age-band groups with SLDs, indicating a deterioration in motor score over time. Their study also, unsurprisingly, revealed that if concomitant SLDs are present in children with DCD, the severity of motor dysfunctions increases until 100% of children score on or below the 15th percentile. However, no data were available on the presence or absence of ADHD in all groups [43]. 

In addition to total M-ABC scores, the three studies that assessed motor performance of SLDs with M-ABC revealed that manual dexterity is the most affected subscale (54.7% [Ibrahim et al., 2019], 52.6% [39] and 34% [43] of children performing on or below the 15th percentile). Regarding the aiming and catching subscale score, the results revealed that 43.2% [19], 40.9% [39] and 24% [43] of the SLD participants performed on or below the 15th percentile scores. The least altered subscore was balancing skills, where 33.8% [19], 33.6% [39] and 27% [43] of the children scored on or below the 15th percentile. 

Moreover, in addition to the M-ABC test, the study of Ibrahim and colleagues also used the Pediatric Evaluation of Disability Inventory-Computer Adaptive Test (PEDI-CAT) for motor SLD evaluations. This test was used to assess the functional mobility of children by parents/caregivers, responding with mobility domain questions regarding the child’s ability to move in different environments and, for instance, perform functional actions. The PEDI-CAT results indicated that most children with SLDs (87.2%) experienced average difficulties in functional mobility [19]. Moreover, this study investigated possible links between M-ABC scores and functional mobility results. The regression analysis showed that both manual dexterity and balance were significant predictors of functional mobility scores in children and adolescents with SLDs [19].

#### 3.1.2. Quantitative Postural Control Assessment in Mixed Group of Learning Disabilities 

The studies of Poblano (2001), as well as the one of Blanchet (2022), investigated quantitative measures regarding postural control. With the Equitest equipment platform, Poblano and their colleagues used the Sensory Organisation Test (participants attempt to maintain a stable, quiet stance in six different sensory conditions) and the Motor Coordination Test (sudden posterior and anterior small, medium and large translations of participants’ support surface) in order to assess the balance skills of 27 Mexican children with SLDs aged between 9 and 10 years old and a control peer group. Intellectual disabilities, ADHD and other neurological signs that possibly included DCD were excluded, but authors did not mention this specific co-occurring disease even if cerebral paralysis, psychiatric disorders and epilepsy were noted. The analysis of mass movements in their study revealed only one significant difference between SLD and control group during the Motor Coordination Test under large translation conditions [40]. Blanchet and colleagues recently (2022) assessed postural control before and during stability limits were reached as well as during maintaining maximal stability limits with an AMTI force platform of 10 SLD Canadian children recruited from special-needs elementary schools. All the participants underwent ADHD and DCD diagnostic assessments. They found significantly lower stability limits and inferior stability in children with SLDs (without ADHD and DCD).

#### 3.1.3. Quantitative Assessment of Fine Motor Skills in Mixed Group of Learning Disabilities

As mentioned in the first results section, manual dexterity is the most affected subscales in the M-ABC test. In Rome, Galli and colleagues (2011) used quantitative assessments of an upwards and downwards drawn spring test for 18 SLD children aged from 8 to 12 years old and a control group of their peers. Children with SLDs drew significantly smaller lines, produced more errors in both lines (1.4 times higher) and angles (2.2 times higher), drew shorter spires, and had greater variability than their peers. Consequently, the crosses they drew had non-uniform side lengths and oblique instead of perpendicular lines. Moreover, 28% of the SLD participants had difficulties or could not perform the downwards spring drawing, while no difficulty was found in the upwards spring test. However, no information was given about DCD or ADHD. 

A variety of qualitative motor tests have been used in studies across countries. Some quantitative motor assessments were also conducted on SLDs. The main result of this section is that all studies found minor or major motor impairments in children with SLDs. The methodology section reveals a large disparity in the samples, but three studies that excluded children with ADHD and DCD [19,23,40] also reported motor impairments, especially in complex motor tasks [40,65,67,80].

Interestingly, the study of Westendorp and colleagues (2011), as well as the study by Vuijk and colleagues (2011), have indicated that this relationship may vary depending on the different areas of academic performance (i.e., reading, spelling, and mathematics) and the kind of motor skill. By using the Test of Gross Motor Development-2, a specific relationship was found between reading abilities and locomotor skills. A trend was found in the relationship between mathematical abilities and object-control skills. They also demonstrated that the larger the children’s learning lag, the poorer their motor skills scores [11]. Regarding M-ABC scores, the children’s spelling and math abilities both had significant effects on the total M-ABC score. Moreover, statistical analysis revealed a significant correlation between the manual dexterity subscale and spelling abilities, the ball skills subscale and reading abilities, and the balance subscale and math abilities. Additionally, it was reported that children with reading problems have a higher risk of fine motor problems [39].

Since all SLDs were included in these samples, the next sections report on studies on a single SLD: dyslexia, dyscalculia and dysorthographia. However, Hussein and colleagues (2020) investigated the association of BOT-2 subtests and participants with particular learning disability types (dyscalculia (*n* = 58), dyslexia (*n* = 22), and mixed (*n* = 20) groups). Their results indicated that there were significant differences among children with different types of SLD in terms of fine motor integration (100% of dyscalculia, 100% of dyslexia and 80% of mixed groups are below the average), bilateral coordination (93% of dyscalculia, 64% of dyslexia and 60% of mixed groups are below the average), balance (48% of dyscalculia, 100% of dyslexia and 40% of mixed group are below the average), and running speed and agility (59% of dyscalculia, 91% of dyslexia, 100% of mixed group are below the average).

### 3.2. Motor Assessment Results in Dyslexia

#### 3.2.1. Standard Qualitative Motor Assessment Batteries in Dyslexia

Some standard motor tests were used for dyslexic children and adolescents, such as M-ABC [44,47,48], TGMD-II [44] and BOT-2 [56]. The research group that used the BOT-2 to assess motor performance of 8-to-11-year-old Brazilian students with dyslexia (no specification in their study regarding ADHD and DCD exclusion) demonstrated that children with dyslexia were statistically poorer than the control group in terms of fine manual control, manual coordination, strength and agility, and total motor composition [56]. Similarly, with the M-ABC measurements conducted on pre-pubertal dyslexic participants aged between 10 and 12 years old from Norway (sample excludes other comorbid diagnoses expected to interfere with motor problems). The results indicated that more than 50% of dyslexic children (severe dyslexia determined by specialist evaluation) and poor readers (teacher selected municipality samples comprising the 5% poorest readers) showed definite motor coordination difficulties at or below the 5th percentile compared to the control group. Both groups showed specific difficulties within the manual dexterity subscales and performed significantly worse than controls within the balance subscales, but this difference was not present for ball skills [48]. These results are supported by McPhiliphs and Sheehy (2004). The poor-readers group of Northern Irish children aged 9–10 years (lowest reading group: bottom 10% word percentile scores with ties resolved by reference to NARA percentile scores) showed a significantly lower mean M-ABC standard score compared with the top 10% reading group (no sample specifications regarding comorbid ADHD and DCD). When the performance of the reading groups is compared with the three motor M-ABC subtests, the results supported the study conducted in Norway, indicating that the top reading group performed significantly better than the low-reading group in terms of manual dexterity and balance subscales but not in terms of ball skills [47]. However, the study of Getchell and colleagues (2007) showed only a significant difference in performance between the American dyslexic group aged from 6 to 11 years old and the control group regarding the total balance subtest of M-ABC, with the dyslexic group showing a poorer performance than control children (no information was given about DCD and ADHD) [44]. A deeper investigation indicated that dyslexic children performed significantly worse in the static balance task, while no significant differences were found in the dynamic balance task. Moreover, although motor control improves with age, their analysis revealed that younger dyslexic children (6–7 years old) performed significantly better on the balance subtest than the older dyslexic group (10 and 11 years old), a result also observed for locomotor score in SLD [12]. However, no significant difference was shown between younger and older groups of domestic children in other subscales.

On the order hand, with the TGMD-II assessment, Getchell and their colleagues (2007) observed no significant differences between the dyslexic and control groups in terms of locomotor skills or object control subtests. Interestingly, unlike many public schools, these schools provided extra time to practice motor skills in physical education classes from 3 to 5 times a week, with opportunities for special instruction for those who may want or need this.

#### 3.2.2. Quantitative Gross Motor Assessment in Dyslexia

Researchers assessed gross motor skills ranging from simple speed, through unimanual sequential movements, to the complex bimanual coordination of adolescents aged from 10 to 16 years old in four groups: (1) dyslexia alone, (2) attention deficit disorder alone, (3) attention deficit disorder and dyslexia, and (4) typically developing peers [45]. The sample excluded neurological or psychiatric conditions and came from a special education school for French- and English-speaking children with learning disabilities in Montreal (Québec, Canada). Their results, acquired with the Leonard Tapping Task, revealed that the dyslexic group made overall significantly less taps and made more sequential errors than the control group. The gross unimanual and bimanual sequential coordination impairments in dyslexic participants were, however, not attributable to motor slowness in responding. In addition, with the experience over the trial, the control group performed an increased number of taps in trial 2 compared to trial 1, while the performance of dyslexic children did not significantly improve with practice [45]. 

An investigation of quantitative locomotor skills in children with dyslexia was conducted by Moe-Nilssen and colleagues (2003). They recruited 18 Norwegian adolescents with dyslexia and their peers aged between 10 and 13 years old (no specifications regarding ADHD and DCD). The participants have to execute a walking test (9-m) at four different self-administered speeds (slow, preferred, fast and very fast) in two conditions (flat floor and an uneven surface of two laminated rubber mats that were arbitrarily padded with circular plates of uneven sizes and thicknesses). The results indicated that when adolescents were requested to walk very fast, the control group walked significantly faster than participants with dyslexia. When groups were compared in terms of normalized speed, the dyslexic group demonstrated significantly higher cadence and shorter steps than the controls [46].

#### 3.2.3. Quantitative Postural Motor Assessment in Dyslexia

As shown by various laboratories with qualitative M-ABC balance subscores [44,47,48], upright postural control is a sensitive variable in dyslexia since difficulties with balance are an enduring feature of dyslexia research. The quantitative postural control assessments in dyslexia support this postulate. Throughout our review of the literature on different types of motor skills in SLDs (mixed group or specific learning disability groups), postural skills in children with dyslexia are clearly the most studied topic, with 71 studies. However, in 2006, a meta-analysis of Rochelle and Talcott, conducted on 15 studies, revealed that the association between dyslexia and balance impairment is most strongly influenced by variables such as IQ and ADHD, rather than reading skills. Moreover, in this review, only one study had apparently screened their sample for co-occurring DCD in dyslexic adults [60]; this precluded a further statistical examination of the influence of DCD variables in their review. Hence, in the next section of our review, we focus on studies published in 2006 and report the inclusion sample criteria (Table 1).

First of all, compared to their peers, dyslexic children and adolescents showed larger postural instabilities during quiet upright standing (static postural control) with eyes open [50,51,52,53,55,58,59,81] in various conditions: fixing an LED placed near to (25 cm) or far from (150 cm) participants, alternating their focus between the far and near LED targets [62], performing eye movements to follow a target that is displayed on one side of a monitor, then disappears and immediately reappears on the opposite side with a frequency of 0.5 Hz [49], read a text naturally [57,61], identify the open character (Landolt reading task) [57], and visually guide horizontal and vertical saccades [61], in a tandem position (heel-to-toe) [50], on compliant mats (4-mm foam, 0.02-m-thick or 0.10-m-thick compliant mat) [46,59] on a stable and unstable platform [52,54,58] and in moving room. These were conducted while the platform remained stationary or oscillated back and forward at frequencies of 0.2 or 0.5 Hz [52,53,54], with or without lightly touching a stationary or moving bar [51,54]. Many variables were sensitive to this motor impairment: larger length, higher trunk accelerations, spectral power indices, body sway magnitude and variability, mean power frequency of the center of pressure (CoP) displacements, length, mean and variance velocity of the CoP displacements, larger standard deviation in the lateral CoP displacements, spatial and temporal postural index, mean velocity of CoP, Romberg Quotient, Wobble (standard deviation of movement for all the three coordinates of 3D space), etc. Furthermore, the postural assessment with the Cerebellar Test paradigm, also named the Dyslexia Screening Test (stand up straight, blindfolded, with feet together and arms along sides; children are pushed in the lower back (opposite the navel) and must try to stay as still as they can) revealed significant postural impairments in children and adolescents with dyslexia [60,66]. Interestingly, a significant correlation was established between the severity of the dyslexic disorders and the CoP surface area [59]. However, the poor postural control performance in dyslexic children and adolescents is not related to lexical and semantic reading requirements [57]. Almost all the studies that demonstrated postural control impairments in dyslexia had rigorously excluded or controlled for comorbid disorders (ADHD and DCD). 

#### 3.2.4. Quantitative Assessments of Fine Motor Skills in Dyslexia

The fine motor skills of dyslexic populations have been shown to be altered and the level of alteration seems to be influenced by task complexity. For example, when dyslexic American adolescents tap in time to an entraining metronome at three prescribed rates by moving the index fingers of both hands in unison, in rhythmical alternation, or in more complex bimanual patterns, they showed significant deficits in timing precision for bimanual tasks that required the integration of asynchronous tapping finger responses, but not when they moved their fingers in unison (sample did not have clinical neurological, organic, or uncorrected sensory deficits) [65]. In 2017, another research group observed fine motor control impairments in students with dyslexia. In comparison to the control group, Canadian children with reading difficulties performed significantly worse on the bead-threading task. In contrast, the performance for the peg-board task was similar in both groups (no information about the exclusion of comorbidities) [64].

In a study conducted in Portugal, four groups (dyslexia, learning disabilities, learning difficulties and typical children) from 7 to 12 years old were submitted to a fine motor function assessment using the Dysgraphia Scale. The results indicated that three groups with learning impairments presented a poorer performance in tests of fingers’ opposition, graphestesia and body imitation than the control group. Both groups with neurological disorders presented the worst performance in most tests when compared to group with learning difficulties and typical children groups. When authors looked at handwriting parameters, they observed that all participants in the learning disabilities groups showed dysgraphics [74]. In the same way, dyslexic Chinese participants aged from 7 to 12 years were assessed with the Chinese Handwriting Assessment Tool (CHAT). They wrote significantly slower, with greater average character size and variation in size, than typical children of the same age group. They also wrote with significantly lower accuracy. From the discriminant analysis, writing speed and accuracy were found to be satisfactory discriminators that could discriminate participants into two groups with a reasonably good classification accuracy of over 70% for every grade [63]. 

In the same way, both fine motor tasks included in the Cerebellar Tests (finger-to-thumb and bead threading) revealed that participants from the United Kingdom, aged from 8 to 12 years old, with dyslexia, showed a significantly poorer performance than the control students. Both ADHD and DCD were included in half of the participants in the sample (Table 1) but they observed and described their impacts on their study. The results revealed that at least part of the discrepancy in motor skills was due to dyslexic individuals with additional disorders (ADHD and/or DCD). This study, therefore, provides partial support for the presence of motor problems in dyslexic children. No evidence was revealed of a causal relationship between motor skills, on the one hand, and phonological and reading skills on the other [60].

### 3.3. Motor Assessments Results in Dyscalculia 

#### Qualitative Standard Motor Assessment Batteries in Dyscalculia 

In typical developmental children, gross skill performance has been positively related to mathematical achievement (Test of Gross Motor Development, speed tasks such as jumping sideways and moving sideways, and one precision task (one leg stand)) and fine skills performance (speed tasks such as threading beads and posting coins), precision tasks (drawing a trail from M-ABC-2), early screening inventory, revised)) [82,83,84]. As mentioned in Section 3.1, Vuijk and colleagues (2011) revealed that participants from 7 to 12 years old with SLD showed significant positive correlations between balance M-ABC subscore and mathematical score. In children with dyscalculia, however, compared to several studies focusing on dyslexia, only two studies have investigated this important topic. These assessed gross motor development. 

The first study [68] investigated the motor skills of Flanders children aged from 7 to 9 years with the M-ABC-2 and the Beery–Buktenica Developmental Test of Visual–Motor Integration (VMI). Their result revealed that children with dyscalculia (ADHD and DCD were unspecified in the study) performed significantly lower compared to age-matched peers on all tests. Furthermore, the researchers conducted a second analysis, in which they compared children with dyscalculia with control children that were one year younger than the dyslexic group. The results indicated that children with dyscalculia also performed significantly lower on all M-ABC tests compared to typical-development children that were one year younger. The correlation revealed that the scores for aiming and catching, balance, VMI copy test and VMI visual perception test were significantly related to procedural calculations [68]. Scores for the M-ABC-2 aiming and catching and balance, VMI copy test and VMI visual perception test were significantly related to procedural calculations. Moreover, manual dexterity (M-ABC-2) and VMI motor coordination tests indicated a marginally significant relation with the number-fact retrieval [68]. 

The second study [38] used BOT-2 with a mixed SLD sample (no specification regarding DCD and ADHD in the study). Nevertheless, researchers investigated the association between BOT-2 subtests of fine motor composite and gross motor composites among participants in three learning disability groups (dyscalculia, dyslexia, and mixed groups). The results for 58 participants with dyscalculia aged from 9 to 13 years revealed that 90% of them had below-average standard scores for the total motor composite BOT-2 score. Moreover, 93% of participants with dyscalculia were below average in terms of their bilateral coordination score, 96.6% had a below-average upper-limb coordination score, 48% a below-average balance score, 57% were below average in terms of running speed and agility, 62% for knee push-ups, 97% for their fine motor precision score, and 100% for their fine motor integration score [38]. To the best of our knowledge, no other study corresponded to our inclusion and exclusion criteria for dyscalculia and dysorthographia. 

In sum, the general results of this scoping review revealed that children, those in late childhood and young adolescents with SLDs have poorer motor skills than their peers. This evidence emphasizes the importance of clinical motor assessments and sensorimotor stimulation. On the other hand, it is widely recognized that children with poor motor skills have a strong tendency to adopt a sedentary lifestyle [28,85,86,87], but what is the state of knowledge about lifestyle behavior and participation in these various activities for SLD individuals? 

### 3.4. Lifestyle Behavior and Physical Fitness in Children and Adolescents with SLD

Evidence indicates that motor competence is associated with perceived competence and multiple aspects of health (i.e., physical activity, cardiorespiratory fitness, muscular strength, muscular endurance, and a healthy weight status) [88] and have shown that children’s weak motor skills were critical factors associated with low levels of physical activity [28,33,85,86,87]. Increasingly, studies in the SLD population show a tendency to adopt sedentary behavior [86,89,90,91]. Moreover, some studies indicated a deterioration in motor abilities over time in the SLD [12,43,44]. This acknowledgement was supported by the relevant review of Kumari and Raj (2016), which shows that children with SLD adopted a less healthy lifestyle than their peers and participate less in physical activities [86]. In the same way, a large study, part of the National Survey of Children’s Health (NSCH) in 2007–2008, comprising 45,897 participants with SLD and ADHD aged from 10 to 17 years, evaluated this sedentary lifestyle using interviews, and indicators of the physical, emotional and behavior of participants. This national study, conducted by the National Center for health statistics in the United States, revealed many important indicators, such as [91]:(1)A total of 73% of participants with an SLD and 39% of participants with an SLD and ADHD were less likely to meet physical activity recommendations than their peers (demographic variables and medication use were controlled).(2)Participants with an SLD have a significantly higher sedentary rate, i.e., nearly two additional hours per day versus participants with only ADHD.(3)Preadolescents and adolescents in the three groups (SLD only, ADHD only and SLD/ADHD) were significantly less likely to meet the recommendations for vigorous physical activity compared to their peers (three days per week or more for a minimum of 30 min), both before and after controlling for demographic variables.(4)Individuals with SLD are 46% more likely to be obese than people with typical development [91].

The adoption of a physically active lifestyle at a young age is very important and influences the lifestyle habits that the individual will have in adulthood. The report by Emerson and Baines (2010) supports these points by reporting that more than 80% of adults with SLDs in the United Kingdom do not meet the government’s physical activity recommendations [90]. Moreover, a study conducted in the United States (Minnesota) measured the level of commitment to non-social activities and social activities, which included many motor and physical activities, using observation grids (10 min observations, 4 sessions) among 78 participating adults with an SLD. Their results highlighted a very low engagement rate in social activities of only 3% [89]. Physical inactivity is recognized as a major health problem worldwide by several organisms, such as the World Health Organization, because evidence has demonstrated its link with poor fitness conditions and poor motor skills, as well as an increased risk of many health problems, such as metabolic syndrome, cardiovascular disease, dyslipidemia and type II diabetes [33,92,93]. Disturbing results revealed that people with SLD have a lower life expectancy and a significantly increased risk of premature death compared to the general population [90,94,95]. Notably, people in the Netherlands with moderate and severe levels of SLDs have a mortality rate that is three times higher than that in the general population when all causes are included [90]. A study has shown that cardiovascular diseases (14–20%) and respiratory diseases (46–52%) are among the leading causes of mortality in adults with SLDs. A total of 22% of people with learning disabilities were under the age of 50 when they died, compared with 9% of the general population [94]. This study supports the study by Cantell and colleagues (2008), conducted in Canada, focusing on people with poor motor skills [96]. Furthermore, this sedentary lifestyle may have a negative impact on individual development and lead to the onset of cascade disorders, such as impairments to the child’s self-image and self-esteem, which are crucial to succeeding in learning situations, as was reported in dyslexia disorders [97]. Considering these interactions between children, their environment, and the task at hand might form a bridge between our different findings but requires deeper investigation.

## 4. Discussion

The purpose of this scoping review was to investigate if individuals with SLDs had motor impairments and examined the possible factors that could interfere with this assumption. The majority of studies that we reviewed demonstrate fine, gross and postural motor impairments. These data were mainly extracted from the available quantitative and qualitative motor test assessments, which were classically used in clinical developmental studies. Thus, one finding of this scoping review reveals that SLD children, late childhood and adolescents have poorer motor skills than their peers. These motor impairments could be exacerbated by the complexity of the motor tasks and/or by the presence of comorbidities such as ADHD and DCD [9,13,43,45,58,60,66]. These results support the sub-objective of this scoping review and highlight a possible link between motor impairments and the sedentary lifestyle behavior in patients with SLDs, which could lead to a deterioration in health [90]. These results are discussed in the following sections.

### 4.1. Motor Control Impairments in Children and Adolescents with SLDs without Comorbidities

One finding of this scoping review is that participants with SLDs have poorer motor skills than their peers. As mentioned by some researchers, SLDs are heterogenous diseases [39,50,98]. In addition, the presence of suspected comorbidities, which has not yet been systematically reported and/or controlled for in the studies, could also increase the observed motor impairments [9]. Nevertheless, although further research needs to be conducted to confirm and specify the motor difficulties that are consistently associated with these populations, especially with SWED and SMLD, our scoping review identified a total of 36 studies that met the inclusion criteria, including 17 studies in which the sample excluded other neurological disorders, such as ADHD and DCD, and 2 studies that observed their impacts on the statistical analyses (identified by ** and *, respectively, in Table 1). All these studies reported moderate-to-severe motor impairments. However, in contrast to the results section, the following discussion section only concerned studies without the presence of suspected SLD comorbidities (references with ** in Table 1) to minimize the interactions or impacts that comorbidities may have regarding the interpretation of SDL motor impairments. 

One major finding of the scoping review revealed that about half of the SLD sample evaluated with the M-ABC revealed that participants showed movement dysfunctions (at or below the 15th percentile rank) [19,48]. Children and adolescents with SLDs displayed significant manual dexterity difficulties [45,48,60,65] as well as clear impairments in balance [41,48,49,50,54,55,57,58,59,60,61,62,66]. Both manual dexterity and balance were also significant predictors for functional mobility in children and adolescents with SLDs [19]. Moreover, the other half of the dyslexic children were within the control range, although they not among the very best performers [60]. Only one study observed no significant difference between Mexican children with SLDs aged from 9 to 10 years and their peers, with one of their two tests aimed at assessing postural control: the Sensory Organization Test paradigm (SOT) [40]. This could be because they included additional recruitment factors in their study, such as an adequate familial environment and middle socioeconomic status; these criteria were more discriminate than those applied in other studies. Further studies and meta-analysis are required to investigate this hypothesis. Nevertheless, when using the Movement Coordination Test, this study revealed differences between groups. Indeed, the timing precision of motor coordination was significantly altered in SLD participants with fast support in platform translation [40] as well as with tapping paradigms [45,65]. For example, participants with SLDs demonstrated significant deficits in bimanual tasks that required the integration of asynchronous tapping finger responses [65], made significantly fewer taps overall and made more sequential errors than the control group [45]. With the experience gained over the trials, the control group performed an increased number of taps in trial 2 compared to trial 1, while the SLD group’s performance did not improve significantly with practice [45]. Another study revealed that 82% of children with dysphasia are classified as “poor” according to the standard scores of the Test of Gross Motor Development [99]. In addition, parents had indicated, using PEDI-CAT, that 87.2% of children with SLDs experienced average difficulties in functional mobility [19]. 

It is possible that the cohorts of children and adolescents with SLDs without previously identified comorbidities comprised subtypes with motor disorders. Interestingly, Ramus and their colleagues (2003) observed that a number of dyslexic children were clearly out of the control range, although it is quite remarkable that the worst performers for one task in the Cerebellar Tests were not the worst performers for the other. This postulate supports the hypotheses of SLD subgroups and highlights the importance of investigations into the fine, gross and postural skills of children with various SLDs. Moreover, some studies indicated that poor motor skills interfere with learning abilities [29,30,31,32,100]. Still, many questions remain regarding the possible origin of SLD motor impairments. Do they have any specific underlying neurological disorders that cause subtypes of motor disorders? Do they have associated motor difficulties that could be remedied by appropriate early interventions? Due to the lack of systematic motor assessments, do these difficulties evolve over time into motor disorders? Do their motor disturbances accentuate the manifestation of learning impairments? These open questions deserve to be asked. The cause-and-effect relationships regarding motor skills in children and adolescents with SLDs must be investigated in each specific learning disorder, such as dyslexia, dyscalculia, and dysorthographia, as well as in other disorders such as dysgraphia and dysphasia. 

### 4.2. Which Motor Mechanisms Are Possibly Altered in SLDs?

In previous decades, some research groups have deepened the understanding of the mechanisms that could potentially be part of these motor impairments using functional and standardized tests as well as neuroimaging techniques [16,17,18]. Some studies observed that certain brain regions in dyslexic participants might be anatomically atypical and exhibit extreme or reversed asymmetry, whereas other brain regions might not differ from controls [98,101,102,103]. Zadina and colleagues (2006) have hypothesized that various combinations of anatomic configurations that were found could explain these dyslexic subgroups depending on the underlying impairments such as phonologic, orthographic, semantic [98]. Our scooping review suggests the addition of SLD motor subtypes in dyslexia. 

The static and dynamic movement parameters are controlled through feedforward and feedback mechanisms based on sensory inputs (visual, tactile, proprioceptive and vestibular), and this is what allows for the efficient control of movement and posture during predictable and unpredictable internal and external perturbations (sensorimotor mechanisms) [23]. Many potential causes of motor impairments have been identified in these different mechanisms through the literature. For example, the persistence of primary reflexes such as the asymmetrical tonic neck reflex (ATNR) could indicate poor neurological development and immaturity within the nervous system [3,47,104]. Other groups of scientists have hypothesized that there is a sensorial process disturbance in SLD. For example, children experience difficulties in dynamically reweighting sensory cues, which are overcome when more informative sensory cues are provided [54], and are not able to compensate with other available inputs when sensorial inputs are less informative (foam or eyes closed) [59]. Some results indicate that the poor motor performance in SLD children is related to how sensory information is acquired from the environment and used to produce motor responses [53]. However, Pozzo and colleagues postulated that sensory deficiencies are not related to short-latency reflexes through feedback regulation, which seemed insufficient to ensure equilibrium, but a higher level of the central nervous system that must integrate all available sensory inputs to construct a global postural estimation and predict body oscillations as well as body scheme [105]. 

Many hypotheses regarding sensorial modality deficits have also been made. Firstly, children and adolescents with SLDs had problems with oculomotor and visuospatial functioning [80,106,107] and could not efficiently reweigh visual cues compared to their peers [54]. They have selective deficits in vibrotactile sensitivity [108] and, recently, proprioceptive deficits have also been demonstrated in children with dyslexia when a passive elbow flexion task was performed by a robotic device [76]. Moreover, Van Hecke’s review indicated that many authors supported the idea of vestibular deficits in children with dyslexia [77]. Therefore, this evidence is compatible with the hypothesis of a generalized, multisensory deficit in temporal and spatial processing functions in dyslexia [108]. The abnormalities of magnocellular stream and cerebellar regions found in dyslexia affected the processing of all sensory systems, and body scheme (internal modeling) might be damaged during early SLD development [81,109,110,111]. Based on a performance index of motor imagery, Marchetti et al. (2022) reported that only dyslexic adults with sensorimotor impairments showed a lower efficiency in terms of mental imagery. In the same vein, Van de Walle de Ghelcke and colleagues (2020) recently claimed that the process of action representation is also impacted in adolescents with developmental dyslexia.

Altogether, these signs could contribute to the development of an accurate internal representation (body scheme) and, consequently, lead to altered motor control, which could lead to automation and initiating movements as well as the postural problems demonstrated in SLDs [13,59,60,81,111,112,113,114]. In fact, the construction of new representations, as well as the re-actualization of previously acquired representations, constitute two distinct mechanisms that are indispensable for learning during ontogeny. The error signal between the predicted consequences of the action and its real sensory consequences allows for the internal representations to be updated, inhibiting the old ones within a time frame that is compatible with the working memory. The development of internal representations of these actions is, thus, linked to the development of executive functions such as anticipation, adaptation, and inhibition [115]. This line of research represents a crucial issue when understanding the improvements that occur from early emergence during childhood [115] to late maturation in late adolescence [16,115], and also for understanding the neurodevelopmental disorders that emerge during these periods [116,117].

Nevertheless, this hypothesis is more strongly supported in dyslexia than in other types of SLD due to the amount of published evidence. The cause of motor impairments in each type of SLD are possibly the different altered motor mechanisms that occur even in the same SLD type, but there could also be similarities. Actually, these hypotheses have been debated due to their methodological shortcomings and uncertainty regarding the presence of comorbidities in some study samples [13,77,118,119].

### 4.3. The Necessity to Investigate the Comorbidity

Our review also supports the necessity of investigating SLD comorbidities. Looking at a total of 34 review studies, our scoping review identified 16 studies in which investigators did not achieve or report an adequate exclusion of comorbid disorders. For example, if DCD was included in the SLD group, which could represent 17.8% of children (based on the Italian reference) [120]. More specifically, in dyslexia, the literature indicates that this coexistence with DCD could occur in from 35 to 50% of cases [56,68,121,122]. Indeed, the comorbidity between SLD and other neurodevelopmental disorders may explain some of the discrepancies and lively debates in the literature. On the other hand, some studies have used SLD exclusion DSM-V criteria diagnostics in their sample, or controlled for the impact of comorbidity with their statistical analyses and showed motor impairments in SLDs. Thus, comorbidities should be reported and investigated to better understand the etiology of SLD and consider the polymorph profiles [13,14,60,117].

In the present paper, we looked at the various methods used to identify and exclude students who may have movement disorders in the reviewed studies. For example, to support their exclusion criteria, some studies conducted in-depth clinical neurological assessments, others looked at the children’s history, and some studies simply did not mention their method of controlling ADHD and DCD sample exclusion. In contrast, other studies had severe protocols and excluded all children with SLDs who had an M-ABC motor performance below the 20th percentile, but the 15th percentile was used as the threshold to identify a motor disorder [13]. Thus, the verification of the methodologies used to identify morbidities is important when interpreting the data and reducing the observed variability in future studies, especially if the participants were recruited from a special needs school, in which the presence of comorbidities is increased. To date, the question remains as to whether children with SLDs who have motor impairments systematically have DCD or a subtype of SLDs with associated motor difficulties, and whether this may explain some of the results observed in the literature. Taken together, the results of our scoping review emphasize the need to increase scientific and clinical investigations of motor disorders in various types of SLD and their comorbidities. Notably, these data support collective thinking about the need for a systematic clinical investigation of motor skills in children with SLD, given the critical impacts that poor motor skills have on their development.

### 4.4. A Relevant Systematic Clinical Investigation of Motor Skills in Children with SLDs as an Early Indicator of Developmental Process Impairments

One of the essential characteristics of motricity is that it allows for a dialog between the individual and their environment. This interaction is possible due to an early coupling that occurs between the perceived environment, both physical and social, and the engagement to act on this environment [123,124]. This early perception–action coupling constitutes the basis of sensorimotor representations [115,116,117]. These sensorimotor representations, as underpinned by the mirror neuron system [125], play a key role in motor control, allowing one to anticipate and act on one’s environment according to the theory of internal models [126,127,128]. Predictive control enables online correction by comparing real sensory feedback with predicted sensory consequences, which are generated through a forward internal model. In the case of a mismatch, error signals allow for the correction of the motor command in real-time. The strength of this functional link between action and perception, highlighted by various neurophysiological studies, is established early in infancy and is the basis of the development of sensorimotor, cognitive, and social representations [115,129]. The identification of motor disorders, as well as poor motor skills, is, therefore, very important, particularly in early development, to prevent impairments in anticipation, adaptation and learning functions [13,84,104,124].

It is well known that poor motor skills in children, such as those observed in our review, are associated with functional mobility impairments, weak physical fitness, the development of low perceived self-efficacy, low participation in daily activities and low motivation to participate in these activities [19,28,33,85,86,87,88,130,131,132,133]. In addition, the degree of locomotor skills and skills related to the control of objects are significantly associated with the practice of physical activity in young children [134,135,136]. Thus, the poor motor skills present in SLDs, which consequently lead to sedentary behavior and social exclusion, support the worrying results revealed in our results section (see Section 3.4). Moreover, motor skills may have an indirect effect on the internalization of problems via factors such as psychological and peer problems [137]. Their association with physical self-perception and autonomous motivation suggests that motor skills also play an important role in the psychological factors that significantly influence children’s participation in physical activity and social inclusion, as well as motor and physical assessments [28,131,132,133]. Therefore, in addition to their functional aspect, motor skills are a critical indicator of children’s development. This supports the relevance of their early systematic clinical investigation.

The dynamic and synergistic role that motor competences play in the decline in participation in physical activities (Stodden Model) induces suboptimal sensorimotor inputs during these important developmental stages and could limit proper brain development. This leads to lifelong consequences; notably, increased motor impairments [21,24]. Thus, the adaptation of recess, social, leisure, recreational, school, physical and motor activities, as well as the environment, are essential to increase regular participation in various types of activities from a young age, especially for children and adolescents with poor motor skills, such as those with SLDs [86]. Therefore, the ecosystem in which this pediatric population participate could benefit from a robust and empirical approach such as a self-determination theory and constraint-led framework [66,67,100,132,133,134,135,136,137,138,139,140,141]. The adapted physical activities and environments, as well as the pedagogical options, are a possible key to promote the participation of children and adolescents with SLDs, which could lead to improvements in sensorimotor parameters. In fact, the systematic review of Barela and colleagues (2016) found physical activity to be a positive correlate of skill composites and motor coordination. In fact, sensorimotor development is influenced by experience refining the sensory and motor functions, as well as the central integration of sensory information [138], especially during the sensitive periods of development [21,22,24,25]. More specifically, in SLDs, the meta-analysis of Rochelle and Talcott (2006) supported these results for SLDs and revealed that balance training can specifically and directly transfer to improvements in the literacy of children with dyslexia. In addition, the review of Kumari and Raj (2016) suggested that regular physical activity promotes overall well-being and academic performance and learning in children with SLDs.

Recently, an increasing number of studies suggest that the use of motor imagery and action observation, either in isolation or combination, are potentially important rehabilitation tools for subjects with internal representation impairments [139]. Better targeting of the sensorimotor contours of SLDs should allow for a more holistic remediation, based on representation improvements, to support children throughout life, thus limiting collateral impacts such as decreased self-esteem and stress in learning or playing situations. In the same vein, developing an assessment tool for internal representations could be further recommended as part of the global assessment of SLDs [116,117].

Even if there are no complaints about the children’s motor skills, early complete motor assessments are essential to optimize development, ensure health care during growth and improve their lifestyle behavior and social inclusion. Moreover, it is recommended that these assessments are continued throughout adolescence, since the probability that motor skills will deteriorate over time is high in children with SLDs [12,43,44] due to their sedentary lifestyle.

## 5. Conclusions

Our scoping review revealed that all of the 36 studies reported motor impairments in SLD students. These motor impairments could be exacerbated by the complexity of the motor tasks and/or by the presence of comorbidities such as ADHD and DCD. Nevertheless, we identified 19 studies in which investigators had controlled or excluded for comorbidities in their SLD samples, and they still demonstrated fine, gross and postural impairments. Further research is needed to confirm and specify the motor difficulties that are consistently associated with these populations and investigate whether SLD motor subtypes are confirmed. It is also probable that the underuse of motor skills at a critical time in development can lead to cascading motor impairments and social interaction problems. This alternative hypothesis could easily be tested by early and appropriate sensorimotor stimulation programs. Investigating correlates of motor skills in children and adolescents with SLDs is an emerging research area with much scope for future investigation. These results highlighted a possible link between motor impairments or motor underuse and the sedentary lifestyle behavior in SLDs, which could lead to deteriorations in health, and supports the need for systematic complete motor assessments in young SLD populations. It is, therefore, essential to increase the state of knowledge on this important topic in the literature and to early investigate sensorimotor skills in SLDs, as several impacts could be underestimated.

## Figures and Tables

**Figure 1 children-09-00892-f001:**
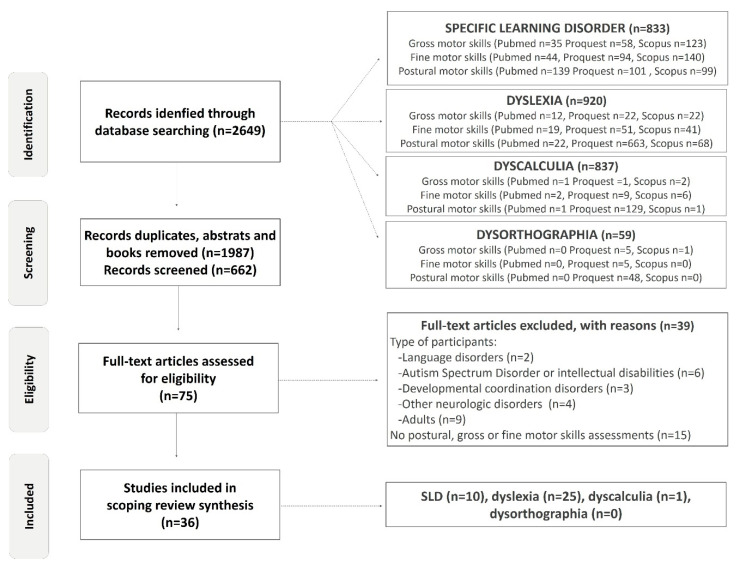
Flow diagram of included studies.

**Table 1 children-09-00892-t001:** Includes sample descriptions, motor tests used, and comorbidity exclusions reported by authors of 36 of the reviewed studies. The ** represents studies that excluded ADHD and other neurologic or motor disorders from their sample, and the * represents studies that included comorbidities but controlled for them in their analysis.

Study	Participants	Motor Tests	Comorbidity Exclusions
**SLD**			
Hussein et al., 2020 [38]	100 students with SLD (dyslexia, dyscalculia, or mixed) 100 typical children - Aged from 9 to 13 years; - Egypt.	Bruininks–Oseretsky test of motor proficiency, second edition (BOT-2).	- IQ - Needed assistance during walking or used walking aids; - Sensory problem such as bing deaf or blind; - DCD and ADHD are unspecified.
Westendorp et al., 2011 [11]	104 students with SLD - Aged from 7 to 12 years; - Netherlands (northern regions).	Test of Gross Motor Development-2	- IQ; - ADHD; - Autism Spectrum Disorders; - DCD is unspecified.
Westendorp et al., 2014 [12]	91 students with SLD 46 control students - Aged from 7 to 11 years; - Netherlands (northern regions).	Test of Gross Motor Development-2	- IQ; - DCD and ADHD are unspecified.
Vuijk et al., 2011 [39]	137 students with SLD, included ADHD (16.8%) and PDD-NOS (13.9%), - Aged from 7 to 12 years; - Netherlands (northern regions).	M-ABC	- IQ; - DCD is unspecified.
** Ibrahim et al., 2019 [19]	148 students with SLD (dyslexia, dyscalculia, dysgraphia and/or slow learner based on Dyslexia Association Malaysian criteria) - Aged from 4 to 16 years; - Malaysia.	- M-ABC; Pediatric Evaluation of Disability Inventory-Computer Adaptive Test (PEDI-CAT).	- Diagnosed with other conditions than SLD; - Needed assistance during walking or used walking aids; - Other motor disabilities, and severe sensory problems, such as being deaf or blind, which affect their ability to learn and perform daily activities; - Other motor disabilities.
** Poblano et al., 2001 [40]	27 students with SLD 27 control students - Age between 9 and 10 years; - Mexico (Mexico city).	- Sensory organization tests; - Movement coordination test.	- IQ; - ADHD; - Epilepsy, cerebral palsy, psychiatric disorders, or other neurologic signs; - Adequate familial environment, middle socioeconomic status; - Congenital malformations; - Phoniatric or visual alterations.
** Blanchet et al., 2022 [41]	74 students with SLD but only 10 without comorbidities were included 20 control students - Age between 8 and 14 years; - Canada (Québec).	Forward, backward, leftward and rightward stability limits with eyes open or eyes closed, standing on foam conditions.	- IQ; - DCD; - ADHD.
Galli et al., 2011 [15]	18 students with SLD 24 control students - Aged from 8 to 12 years; - Italia (Rome).	Upwards and downwards spring test	- IQ; - DCD and ADHD are unspecified.
** Okuda and Pinheiroa 2015 [42]	10 students with learning difficulties, 10 control students - Aged from 8 to 11 years; - Brazil.	Bruininks–Oseretsky Test of Motor Proficiency	- Pre-natal, peri-natal and post-natal complications - Neuropsychomotor development and language delays (phonoaudiological, neurological, educational, occupational therapy, and neuropsychological assessments).
Jongmans 2003 [43]	94 SLD students without DCD, 57 students with DCD but without SLD, 53 students with both DCD and SLD, 545 control students (without DCD and without SLD) - Aged from 4 to 13 years; - Netherlands and Germany.	M-ABC	- IQ; - Physical or sensory disorders; - DCD (DSM-IV diagnostic) for the SLD group; - No data were available on the presence or absence of behavioral or conduct disorders, especially ADHD.
**DYSLEXIA**			
Getchell et al., 2007 [44]	26 students with dyslexia - Younger dyslexic group (*n* = 7; age range: 6 to 7 years); - Older dyslexic group (*n* = 7; age range: 10 to 11 years). - 23 control students - United States (New Castle County).	- M-ABC - Test of Gross Motor Development	- Potentially undiagnosed learning disability, 14-item Learning Disability Checklist (Haler 2005); - DCD and ADHD are unspecified.
** Marchand-Krynski et al., 2017 [45]	27 students with dyslexia 27 students with attention deficit disorder with or without hyperactivity disorder (AD) student group 27 students with Dylexia + AD 27 control students - Aged from 10 to 15 years; - Canada (Québec).	The Leonard Tapping Task	- IQ; - Neurological or psychiatric conditions; - Dysphasia and dyscalculia; - Traumatic brain injury.
Moe-Nilssen et al., 2003 [46]	18 students with dyslexia 21 control students - Aged from 10 to 13 years; - Norway.	Standing on firm, compliant mat (0.00-m-thick) and compliant pillow (0.10-m-thick) during quiet standing (eyes open (EO), with eyes closed (EC), and walking.	- IQ; - DCD and ADHD are unspecified.
McPhillips and Sheehy, 2004 [47]	41 poor readers students (bottom 10% based on WORD percentile scores with ties resolved by reference to NARA percentile scores) 41 middle reader students (middle 10%) 41 good reader students (top 10%) - Aged from 9 to 10 years; - United Kingdom, Northern Ireland.	- M-ABC - Asymmetrical Tonic - Neck Reflex (ATNR)	- IQ; - DCD and ADHD are unspecified.
** Iversen et al., 2005 [48]	20 students with dyslexia (severe dyslexia referred to specialist evaluation) 17 poor readers (teacher selected municipality sample comprising the 5% poorest readers) 22 good readers (control group) (teacher selected municipality sample comprising the 5% best readers) - Aged from 10 to 12 years; - Norway.	M-ABC	- IQ; - ADHD; - Tourette; - Other comorbidities expected to interfere with motor problems.
** Barela et al., 2020 [49]	12 students with dyslexia 12 control students - Aged from 9 to 12.3 years; - Brazil.	Quiet upright stance in both fixation and guided conditions (fixate on a target that appeared and disappeared on the left and right sides of the monitor). Body sway was measured with OPTOTRAK. Eye movements were tracked using eye-tracking glasses (ETG 2.0-SMI).	- Complete evaluation and dyslexia screening assessment including neurological, psychological, and phonological abilities.
* Kaplan et al., 1998 [37]	224 students with learning reading disorders/attention disorders. 155 control students - Aged from 10 to 14 years; - Canada (Alberta).	- M-ABC; - Bruininks–Oseretsky Test of Motor Proficiency; - DCD Questionnaire completed by parents.	- IQ; - Problems with motor skills.
** Brookes 2010 [50]	16 students with dyslexia 24 control students 17 adults with dyslexia 30 control adults, - Students aged from 11 to 14 years; - United Kingdom (London).	- Heel-to-toe quiet standing balance (1 min); - Stand on a mat in eyes-open and eyes-closed conditions.	- IQ; - ADHD; - Additional diagnoses of developmental disorders.
Viana et al., 2013 [51]	30 students with dyslexia 30 control students - Aged from 9 to 12 years; - Brazil (São Paulo).	Quiet standing balance inside a moving room under five conditions: (1) no vision and no touch; (2) vision moving room; (3) vision moving bar; (4) vision moving room and stationary touch; (5) vision stationary room and moving bar.	- IQ; - Hyperactivity; - NO specification about other developmental disorders.
Barela 2011 [52]	10 students with dyslexia 10 control students - Aged from 10 to 11 years; - Brazil.	Quiet upright standing balance inside a moving room that remained stationary or oscillated back and forward at frequencies of 0.2 or 0.5 Hz.	- DCD and ADHD are unspecified
Razuk et Barela, 2014 [53]	18 students with dyslexia 18 control students - Aged from 9 to 13 years; - Brazil.	Quiet upright standing balance inside a moving room; looked at a target at different distances between the participant and a moving room frontal wall (25–150 cm) and with different vision (full and central).	- DCD and ADHD are unspecified
** Razuk et al., 2020 [54]	20 students with dyslexia 19 control students - Aged from 10 to 11 years; - Brazil.	Quiet upright standing balance inside a moving room and another balance test with OR during lightly touching a moving bar with three different stimulus characteristics: low (pre-transition), high (transition), and low-amplitude (post-transition).	- DSM-5 classification; - Orthopedic, neurological, and musculoskeletal conditions.
** Goulème et al., 2015 [55]	30 students with dyslexia 30 control students - Aged from 9 and 10 years; - France (Paris).	Quiet upright standing balance was evaluated using Multitest Equilibre from Framiral^®^. Posture with eyes open and eyes closed under stable and unstable platform conditions.	- IQ; - Drug treatment, orthopedical, neurological, psychological ophthalmological and phonological abnormalities (pediatric hospital evaluation).
Okuda et al., 2014 [56]	19 students with dyslexia 60 control students - Aged from 8 to 11 years; - Brazil.	Bruininks–Oseretsky Test of Motor Proficiency (second edition)	- DCD and ADHD are unspecified.
** Razuk et al., 2018 [57]	15 students with dyslexia 15 control students - Aged from 8 to 11 years; - France (Paris).	Quiet upright standing balance during text reading and Landolt reading. Eye movements (Mobile T2^®^, SuriCog) and center of pressure excursions (Multitest Equilibre)	- IQ; - Extensive examination, including neurological, psychological and phonological abilities; - All students underwent ophthalmologic/orthopedic examinations for visual, sensorial and motor functions.
** Bucci et al., 2017 [58]	23 students with dyslexia 23 students with Autism Spectrum Disorder (ASD) 23 students with Attention deficit/hyperactive disorder (ADHD) 23 control students - 10 years old; - France (Paris).	Quiet upright standing balance (Multitest Equilibre) on both a stable and an unstable platform, under two different visual conditions: eyes open and eyes closed.	- IQ; - DCD. - The comorbidities of ASD and ADHD; - Comorbid ADHD and dyslexia.
** Goulème et al., 2017 [59]	24 students with dyslexia 24 control students - Aged from 6 to 11 years; - France (Paris).	Quiet upright standing balance (TechnoConcept^®^ platform) on both a firm and foam support surface under two different visual conditions: eyes open and eyes closed	- IQ; - Drug treatment or orthopedic anomaly; - Complete, including neurological, psychological and phonological abilities.
* Ramus et al., 2003 [60]	22 students with dyslexia (included 7 with ADHD, 1 with DCD, and 2 both ADHD and DCD) 22 control students - Aged from 8 to 12 years; - United Kingdom (London).	Cerebellar tests	- IQ; - ADHD; - DCD; - Basic auditory dysfunction.
** Legrand et al., 2012 [61]	18 students with dyslexia 18 control students - Aged from 9 to 11; - France (Paris).	Quiet upright standing balance (TechnoConcept^®^ platform) during visually guided saccade task and silent reading of a text.	- IQ; - Signs of hyperactivity; - Signs of DCD; - Extensive examination; - Included neurological, psychological and phonological abilities.
** Kapoula et Bucci, 2007 [62]	13 students with dyslexia 13 control students - Aged from 10 and 14; - France (Paris).	Quiet upright standing balance during fixing LED placed near to (25 cm) or far from (150 cm) participants with fixation alternated between the far and the near LED targets.	- IQ; - Extensive examination, including neurological, psychological and phonological tests.
** Lam et al., 2011 [63]	137 students with dyslexia 756 control students - Aged from 7 to 12 years; - China (Hong Kong).	Chinese Handwriting Assessment Tool (CHAT)	- IQ; - Medical or physical disabilities.
Niechwiej-Szwedo et al., 2017 [64]	19 poor readers (who were reading below expected grades and age-level) 19 typically developing children. - Aged from 5 to 11 years; - Canada (Ontario).	Bead-threading and pegboard	- DCD and ADHD are unspecified.
** Wolff et al., 1990 [65]	50 students with dyslexia 50 control students 40 dyslexic adults - Students aged from 13 to 18 years; - The United States (Boston).	Tap in time to an entraining metronome at 3 prescribed rates by moving the index fingers of both hands in unison, in rhythmical alternation, or in more complex bimanual patterns.	- IQ; - Clinically significant neurological, organic, or uncorrected sensory deficits; - Not exposed to an adequate learning environment before enrolling in the residential school.
** Naz and Najam 2019 [66]	24 reading disorder students (pure), 24 reading disorder + ADHD students 24 control students - Students aged from 11 to 15 years; - Pakistan.	- Rey Osterrieth Complex Figure Task (visuoconstructional ability); - Postural Stability subtest of Dyslexia Screening Instrument; - Dichotic Listening Words Test.	- IQ; - ADHD; - Multidisciplinary clinical diagnostic assessment comprising semi-structured clinical diagnostic interviews with parents and teachers; the Bangor Dyslexia Test has been used widely to record overall neuropsychological assessments; - Used the DSM-V criteria to externalize and internalize disorders of childhood; any evidence of neurological dysfunction, poor physical health, uncorrected sensory impairments, or history or current presentation of psychosis led to exclusion from the study.
Haouès et al., 2021 [67]	30 students with dyslexia (28.9% had ADHD) 30 control students - Students aged from 11 to 15 years; - Tunisia.	Quiet upright standing balance (RunTime^®^) bipedal and unipedal (eyes open and eyes closed).	- Body mass index <30 kg/m^2^; - Proprioceptive treatments; - Neurologic, psychiatric and genetic disorders; - Strabismus.
**Dyscalculia**			
Pieters et al., 2012 [68]	39 students with mathematical learning disabilities, 106 control students. - Aged from 7 to 9 years; - Belgium (Flanders).	- M-ABC-2; - Beery–Buktenica Developmental Test of Visual–Motor Integration.	- IQ; - Native language different from Dutch; - No specification in dyscalculia group of ADHD or DCD.

## Data Availability

Not applicable.

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
