# Peer review of "Specific Learning Disorder in Children and Adolescents, a Scoping Review on Motor Impairments and Their Potential Impacts"

_children, 2022, doi:10.3390/children9060892_

Round 1

Reviewer 1 Report

Comments:

Line 23-24: Do the authors equate motor skills with sensorimotor impairments? This is not necessary similar (but related) concepts?

Line 24: Unclear sentence, what is referred to as ‘complexity of the motor’?

Line 25-27: This sounds like an important point for discussion, as SLD might be associated with poorer motor skills in children due to their lack of motor experience, not necessary a direct link/comorbidity with SLD.

Line 78-93: This overview of important background literature seems to bi a bit superficial. There seems to be around 15 studies cited, but very little information is synthesized and presented from these studies. As this provides (perhaps) the most important background for the review (proceeding from what we know to what we don’t know), it could provide important argumentations for conducting the review. It also seems to be a bit non-logical that the authors aim to summarize studies on motor difficulties while stating at the same time that ‘motor difficulties in these populations remain unknown’ (line 90). Also, the authors suddenly apply the term milestones in line 92. The application of various terminology such as motor skills, sensorimotor impairments, motor deficits, motor milestones, introduce unclarity as to what is the purpose of the review to summarize (see also comment above).

Line 118-119: yet another terminology, fine gross and balance skills. Needs to be connected with rest of the introduction.

Line 119-122: There are conceptual models (such as that of Stodden) that links motor development with physical activity/fitness. This could provide an important framework for the review, as it stresses the relationship between these factors.

Line 135: It is stated that PRISMA guidelines have been followed, but his is clearly not the case. There are many items missing from this guideline. Please align with the stated guidelines and revise. E.g., information is lacking in terms of what information was systematically extracted from the different studies and the PRISMA flowchart is also missing.

Line 157-158: How can it be justified that only two databases were searched? The international recommendation for systematic reviews (e.g., Cochrane, PRISMA) recommends searchikng multiple bibliographic sources. This especially important as the review targets medical research, in which has a tremendous amount of journals. 

Results section:

I must admit that I found the results section somewhat hard to read. The authors jump from result to result, and there are no visual aids or any form of tabulation that can provide the reader of an overview of the results. In some paragraphs, it almost seems like the authors are discussing the different results, rather than providing a systematic overview/analysis of the various study results. Without any systematic overview of the extracted results study-by-study in the form of a table/figure, the results are presented with a mixture of different studies, disabilities, motor assessments etc. in which are hard to comprehend. For a start, a table might present the percentage of children with disability that score below a certain threshold/percentile (dependent upon the test).

Line 661: The manuscript still lacks a proper definition of motor deficits, and what information the authors extract from the various studies to determine this.

Line 664: Unclear what is the meaning of the term “late children’

Line 684: Is there a table 2?

Line 687-710: This appears to me as a repetition of information already provided in the results section. The authors raise a series of relevant questions (720-725) with regard to results from the review, however these are not followed up by empirical and theoretical discussion.

Liner 741: There are other possibilities that could be discussed rather then simply adding subtypes, such as motor deficits emerging from different/lack of relevant environmental exposures.

Line 743-771: Although sensory impairments might have a role in terms of poorer motor skills, this is not discussed by the authors. A sensory impairment does not necessary means poor motor skills?

Line 835: The approach from an environmental perspective seems to be relevant, and I recommend the authors to take into account other environmental/ecological approaches such as the constraint-led framework to explain their findings. Considering the interaction between children, their environment, and the task at hand might provide a bridge between the different findings.

Line 870: As the authors here seems to refer to the Stodden, this model (and other theoretical perspectives presented in the discussion), should be presented in the introduction as a part of the background and rationale for the review.

Author Response

[Children] Manuscript ID: children-1737439

Responses to comments and suggestions for authors

We thank the reviewer for all of their constructive comments. Any revisions to the manuscript are marked and highlighted (addition and deleted text). English needs to be extensively revised but that due to the short deadline, our reviewer was not available. Note that English correction will be done.

Reviewer 1

Line 23-24: Do the authors equate motor skills with sensorimotor impairments? This is not necessary similar (but related) concepts? 

We agree with the reviewer. The term “sensorimotor” can lead to some confusion in the reader's interpretations. We have revised the context in which this term was used in the manuscript and removed it at lines 671, 686, 740, 803, 925.

Line 24: Unclear sentence, what is referred to as ‘complexity of the motor’?

We suggest “these motor impairments are exacerbated by the complexity of the motor activities and the presence of comorbidities” (Line 24-25).

Line 25-27: This sounds like an important point for discussion, as SLD might be associated with poorer motor skills in children due to their lack of motor experience, not necessary a direct link/comorbidity with SLD.

Absolutely, we added this idea clearly at the beginning of the manuscript (line 27) rather than just discussing about it in the discussion section, thanks to the reviewer. We have also improved the conclusion to this effect (lines 938-944).

“These results support our sub-objective and highlighted the link between motor impairments and the sedentary lifestyle behavior of SLD that consequently could lead to health and motor deterioration, due to their lack of motor experience, not necessarily a comorbidity” (line 27).

“It is probable that underuse of motor skills at a critical time in development can also lead to cascading motor impairments and social interaction problems. This alternative hypothesis could be easily tested by early and appropriate sensorimotor stimulation programs. The investigating correlates of motor skills in children and adolescents with SLD is an emerging area with much scope for future investigation. These results highlighted the possible link between motor impairments or motor underuse and the sedentary lifestyle behavior of SLD that consequently could lead to health deterioration and support the need for systematic complete motor assessments in young SLD populations” (lines 940-948).

Line 78-93: This overview of important background literature seems a bit superficial. There seems to be around 15 studies cited, but very little information is synthesized and presented from these studies. As this provides (perhaps) the most important background for the review (proceeding from what we know to what we don’t know), it could provide important argumentations for conducting the review. It also seems to be a bit non-logical that the authors aim to summarize studies on motor difficulties while stating at the same time that ‘motor difficulties in these populations remain unknown’ (line 90). Also, the authors suddenly apply the term milestones in line 92. The application of various terminology such as motor skills, sensorimotor impairments, motor deficits, motor milestones, introduce unclarity as to what is the purpose of the review to summarize (see also comment above).

We had revised all of the reviewer comments. As address in the comment above, we revised the terms used in order to avoid unclarity. In addition to the corrections made in regard of the term sensorimotor, we have also revised and modified various terms in order to unify the text (for example lines 17, 79, 86, 93, 530, 668, 687, 742, 933 and title). In the first version of the manuscript, we found 34 studies who met the inclusion criteria that included 19 studies in which investigators had excluded or control comorbidities. In this section (Lines 78-93), we are only introducing the scoping review topic and their importance. In the results section, all motor impairments were described. We have indeed mistranslated the term “méconnu” with “unknown” rather than “misunderstood.” Thereby, we suggest in lines 89-90: Nevertheless, motor difficulties in these populations remain misunderstood and the question is still open.

Line 118-119: yet another terminology, fine gross and balance skills. Needs to be connected with rest of the introduction.

We thank the reviewer for this observation. We have modified line 78 in order to define that motor impairments in our review will refer to the postural, gross and fine motor skills.

Line 119-122: There are conceptual models (such as that of Stodden) that links motor development with physical activity/fitness. This could provide an important framework for the review, as it stresses the relationship between these factors.

Absolutely, as suggested by the reviewer, we had introduced this conceptual model earlier in the text. Lines 122-123: “Indeed, Stodden and colleagues (2014) proposed a conceptual model that links motor development and Health-Related Fitness during ontogenesis.”

Line 135: It is stated that PRISMA guidelines have been followed, but his is clearly not the case. There are many items missing from this guideline. Please align with the stated guidelines and revise. E.g., information is lacking in terms of what information was systematically extracted from the different studies and the PRISMA flowchart is also missing. Line 157-158: How can it be justified that only two databases were searched? The international recommendation for systematic reviews (e.g., Cochrane, PRISMA) recommends searching multiple bibliographic sources. This especially important as the review targets medical research, in which has a tremendous number of journals.

We are sorry for omitting to attach our Flowchart. It is attached to our second submitted version. We revised all reference section of the studies included in our review in order to minimize the possibility of missing a study. We revised and improved our protocol/methodology section. We add the SCOPUS databases in order to increase the database search for our scoping review. The results of our research are included in the Flowchart. These procedures have made it possible to identify two additional references. However, both of these references were excluded in the first tour:

Haouès et al. 2021 : entitled “case study” so it was excluded in the first round but, we read the protocol and they conducted typical study (Peer-reviewed secondary study).

Naz and Najam 2019: assessed visuoconstructional ability by scores on Rey Osterrieth Complex Figure Task (RCFT) and Dyslexia Screening Instrument. Our student omitted the unusual subtest of postural control in Dyslexia Screening Instrument which usually refers to the reading and writing test.

Results section:

I must admit that I found the results section somewhat hard to read. The authors jump from result to result, and there are no visual aids or any form of tabulation that can provide the reader of an overview of the results. In some paragraphs, it almost seems like the authors are discussing the different results, rather than providing a systematic overview/analysis of the various study results. Without any systematic overview of the extracted results study-by-study in the form of a table/figure, the results are presented with a mixture of different studies, disabilities, motor assessments etc. in which are hard to comprehend. For a start, a table might present the percentage of children with disability that score below a certain threshold/percentile (dependent upon the test).

Line 661: The manuscript still lacks a proper definition of motor deficits, and what information the authors extract from the various studies to determine this.

To answer this question, we add the following sentence: The majority of the studies that we have reviewed demonstrate fine, gross and postural motor impairments. These data were mainly extracted from the available quantitative and qualitative motor test assessments, classically used in clinical developmental studies.

We tried different literary style in order to facilitated the reading. Finally, because many motor task assessments were used (all listed in lines 221 to 240) with their own deficits parameters/ threshold as well as several study designs were conducted was largely inconsistent (for example, sometime authors have compared their results to a gold standard and sometime to children peers), we choose to explain these factors in the description of the results. Thus, before the presentation of each founded result, a contextualization of the sample and motor evaluations is presented in order to optimize the interpretation of the scoping review results by the reader.    

The comments expressed in this section and the question posed later in the text by reviewers are fair with regard to a systematic review. However, due to the nature and the actual confusion about motor impairments in SDL, we had voluntarily sectioned the scoping review. In general, scoping reviews are commonly used for ‘reconnaissance’ – to clarify working definitions or conceptual boundaries of a topic or field. Scoping reviews are therefore of particular use when a body of literature has not yet been comprehensively reviewed, or exhibits a large, complex, or heterogeneous nature not amenable to a more precise systematic review (Peters et al. 2015) such as the SDL. While scoping reviews may be conducted to determine the value and probable scope of a full systematic review, they may also be undertaken as exercises in and of themselves to summarize and disseminate research findings, to identify research gaps, and to make recommendations for future research (Peters et al. 2015, Munn et al. 2018). True to their name, scoping reviews are an ideal tool to determine the scope or coverage of a body of literature on a given topic and give clear indication of the volume of literature and studies available as well as an overview (broad or detailed) of its focus (Munn et al. 2018). To find out whether starting work on one is a good idea, some researchers conduct a scoping review first to find out more about the body of evidence in a particular topic area. Scoping reviews are exploratory, and they typically address a broad question. Researchers conduct them to assess the extent of the available evidence, to organize it into groups and to highlight gaps. Sometimes scoping reviews are also used to decide whether or not it would be useful to conduct a systematic review (www.covidence.org). Thus, scoping reviews serve to synthesize evidence and assess the scope of literature on a topic. Among other objectives, scoping reviews help determine whether a systematic review of the literature is warranted (Tricco et al. 2018). In this light, we decide to conduct a scoping review. We apologize for this confusion. We hope that our corrections will satisfy the reviewer.

Tricco, AC, Lillie, E, Zarin, W, O'Brien, KK, Colquhoun, H, Levac, D, Moher, D, Peters, MD, Horsley, T, Weeks, L, Hempel, S et al. PRISMA extension for scoping reviews (PRISMA-ScR): checklist and explanation. Ann Intern Med. 2018,169(7):467-473. doi:10.7326/M18-0850.

Zachary Munn, Micah D. J. Peters, Cindy Stern, Catalin Tufanaru, Alexa McArthur & Edoardo Aromataris. Systematic review or scoping review? Guidance for authors when choosing between a systematic or scoping review approach. BMC Medical Research Methodology volume 18, Article number: 143.

Micah D J Peters, Christina M Godfrey, Hanan Khalil, Patricia McInerney, Deborah Parker, Cassia Baldini Soares. Guidance for conducting systematic scoping reviews. Int J Evid Based Health.  2015 Sep;13(3):141-6.

http://prisma-statement.org/Extensions/ScopingReviews

Line 664: Unclear what is the meaning of the term “late children’

This is possibly due to a typing error. Late childhood refers to children aged 9-11 years before the onset of puberty. We have made the corrections.

Line 684: Is there a table 2?

We have changed the sentence structure to avoid this type of confusion, lines 728-730. A total of 36 studies that met the inclusion criteria, including 17 studies in which sample excluded other neurologic disorders than SLD such as ADHD and DCD and 2 studies that observed their impacts in the statistical analyses (identified by ** and * respectively in the table 1)”.

Line 687-710: This appears to me as a repetition of information already provided in the results section.

In light to this reviewer comment and the proofreading, indeed, this section appears redundant. Thus, we added a sentence to clarify the main idea of ​​this section because it was directly related to the objective of our scoping review: identified and resumed motor impairments in children and adolescents exclusively WITHOUT the presence of suspected SLD comorbidity.

 “However, in contrast to the results section, the next discussion section only concerned the studies without the presence of suspected SLD comorbidity (references with ** in table 1) in order to minimized the interactions or the impacts of comorbidities on SDL motor impairments interpretation.” (Lines 731-735)

The authors raise a series of relevant questions (720-725) with regard to results from the review, however these are not followed up by empirical and theoretical discussion. Line 741: There are other possibilities that could be discussed rather then simply adding subtypes, such as motor deficits emerging from different/lack of relevant environmental exposures.

We agree with the reviewer. But actually, that was not the purpose of this scoping review, probably for the purpose of a future review! We have nevertheless addressed this point lines 974-977 and we added a sentence in the conclusion. Also, we addressed the interaction between sense, motricity and environment in 4.4 section

Line 743-771: Although sensory impairments might have a role in terms of poorer motor skills, this is not discussed by the authors. A sensory impairment does not necessary means poor motor skills?

We added a sentence in order to explain the link between sensory systems and motor control (lines 796-800). We also discussed about the contribution of sense of internal representation development. However, our manuscript already contains 32 pages. This will be also a purpose of a future review.

Line 835: The approach from an environmental perspective seems to be relevant, and I recommend the authors to take into account other environmental/ecological approaches such as the constraint-led framework to explain their findings. Considering the interaction between children, their environment, and the task at hand might provide a bridge between the different findings.

We agree with the reviewer and this approach seems very relevant to us. Prior to the submission of the first manuscript version, we deleted a section on self-determination theory which dealt with this topic since the manuscript was too long. We had chosen to discuss about this important interaction between sensorimotor children’s skills and their environment in the 934-936. We had a sentence in lines 700-701 and, as suggested by the reviewer, we had briefly introduced the constraint-led framework. Moreover, in regard to previous comment, we already modify the manuscript (line 27, lines 940-943).  

Line 870: As the authors here seems to refer to the Stodden, this model (and other theoretical perspectives presented in the discussion), should be presented in the introduction as a part of the background and rationale for the review.

We suggested the revision made previously according to the reviewer comment above : Line 123.

Reviewer 2 Report

I thank the editor for the opportunity to review the article 'Specific learning disorder in children and adolescents, a scoping review on motor deficits and their potential impacts'. The topic is interesting and the article could be a good contribution in the field of SLD. the article sound in scientifically way but I prefer to see the PRISMA in order to understand the paper selection you have done. the corpus of references are appropriate for the purpose of the article. English is not appropriate and it needs of an extensive revision.

Author Response

[Children] Manuscript ID: children-1737439

Responses to comments and suggestions for authors

We thank the reviewer for all of their constructive comments. Any revisions to the manuscript are marked and highlighted (addition and deleted text). English needs to be extensively revised but that due to the short deadline, our reviewer was not available. Note that English correction will be done.

Reviewer 2

I thank the editor for the opportunity to review the article 'Specific learning disorder in children and adolescents, a scoping review on motor deficits and their potential impacts'. The topic is interesting and the article could be a good contribution in the field of SLD. the article sound in scientifically way but I prefer to see the PRISMA in order to understand the paper selection you have done. the corpus of references are appropriate for the purpose of the article. English is not appropriate and it needs of an extensive revision.

We thank the reviewer. We agree that the English needs to be revised but that due to the short deadline our reviewer was not available. However, while reviewers analyze our proposals in response to their comments, the English corrections will be done. A figure Flow diagram had been included in our scoping review and the methodology was enhanced.

Reviewer 3 Report

Excellent study that provides information on motor deficits and their possible impacts.

It seems to me that the review is correctly written and provides new evidence on the subject of study.

With the aim of clarifying some remaining doubts, I recommend that you add the search strategy used more clearly. It would also be good to add the flow chart, in order to know what was found at first, and what was left as the inclusion and exclusion criteria were met.

I would also request to add citation from WHO, line 124.

Thanks for the work you did, it's very good. Congratulations.

Author Response

[Children] Manuscript ID: children-1737439

Comments and Suggestions for Authors

We thank the reviewer for all of their constructive comments. Any revisions to the manuscript are marked and highlighted (addition and deleted text). English needs to be extensively revised but that due to the short deadline, our reviewer was not available. Note that English correction will be done.

Reviewer 3

Excellent study that provides information on motor deficits and their possible impacts.

It seems to me that the review is correctly written and provides new evidence on the subject of study.

With the aim of clarifying some remaining doubts, I recommend that you add the search strategy used more clearly. It would also be good to add the flow chart, in order to know what was found at first, and what was left as the inclusion and exclusion criteria were met. I would also request to add citation from WHO, line 124. Thanks for the work you did, it's very good. Congratulations.

We thank the reviewer and we are agreeing that the search strategy would benefit from being clarified. We made de correction accordingly and we attach a flow chart in our study.

We had the citation from WHO: “Physical inactivity is one of the leading risk factors for noncommunicable diseases and death worldwide” (line 128). We also add website reference in the reference section (144. World Health Organization. Available online: https://www.who.int/fr (27-05-2022).

Round 2

Reviewer 1 Report

Congratulations, and thank you for taking my comments into consideration.